# In-Depth Analysis of Caesarean Section Rate in the Largest Secondary Care-Level Maternity Hospital in Latvia

**DOI:** 10.3390/jcm12196426

**Published:** 2023-10-09

**Authors:** Laura Racene, Zane Rostoka, Liva Kise, Justina Kacerauskiene, Dace Rezeberga

**Affiliations:** 1Department of Obstetrics and Gynaecology, Rīga Stradiņš University, LV-1007 Riga, Latvia; zane.rostoka@rsu.lv (Z.R.); liva.kise@rdn.lv (L.K.); dace.rezeberga@rsu.lv (D.R.); 2Riga Maternity Hospital, LV-1013 Riga, Latvia; 3Department of Obstetrics and Gynaecology, Lithuanian University of Health Sciences, 50167 Kaunas, Lithuania; justina.kacerauskiene@lsmuni.lt; 4Riga East Clinical University Hospital, LV-1038 Riga, Latvia

**Keywords:** audit, Robson classification, caesarean section, rate, caesarean birth

## Abstract

There is no surgical intervention without risk. A high rate of caesarean sections (CSs) impacts on maternal and newborn mortality and morbidity. For optimisation of the CS rate, regular monitoring is necessary. In 2015, the World Health Organization recommended the Robson classification as a global standard for assessing, monitoring, and comparing CS rates. We analysed all births in 2019 in the Riga Maternity Hospital—a secondary-level monodisciplinary perinatal care hospital in Latvia—according to the Robson classification, seeking to identify which groups make the biggest contribution to the overall CS rate. In total, 5835 women were included. The overall CS rate was 21.5%. In our study, the largest contributors to the overall CS rate were as follows: Group 5 (33.3%); Group 2 (20.8%); and Group 1 (15.6%). The results of our deeper analysis of individual groups (Group 1 and 5) from our study may help to develop targeted interventions for specific subgroups of the obstetric population, effectively reducing both the overall rate of CS and the number of unnecessary CSs performed. The CS rate reduction strategy should be based on decreasing CSs in Group 1 and encouraging VBAC, thus decreasing the number of women undergoing two or more CSs in future.

## 1. Introduction

Increasing rates of caesarean sections (CSs) have become a significant concern for maternal and newborn health globally. Latvia, like many other countries, has also experienced a rise in the CS rate over the past few decades. The CS rate is currently around 21–23% [1]. The high rate of CS raises questions about the quality of obstetric care and the appropriateness of clinical practices in maternity hospitals.

In 2015, the World Health Organization (WHO) recommended the Robson classification as a global standard for assessing, monitoring, and comparing CS rates [2]. This system is designed to classify women into ten groups based on parity, previous CS, onset of labour, number of foetuses, gestational age, and foetal lie and presentation [3]. By using the Robson classification system, healthcare providers can identify groups with the highest operative delivery rate and design multilayered interventions to reach target groups.

This descriptive study offers a detailed analysis of the main reasons for CSs according to the Robson classification in the Riga Maternity Hospital (RMH), a secondary care-level maternity hospital and the largest childbirth institution in Latvia. As more than 30% of parturients in Latvia receive obstetric care in the RMH, changes in the CS rate in the RMH have a noticeable impact on the total CS in rate Latvia. This study aims to identify which groups make the most significant contribution to the overall CS rate and identify specific obstetric goals of the RMH’s CS rate reduction strategy. As there have been many previous discussions about the importance of reducing the CS rate in nulliparous women with spontaneous onset of labour [4,5], we analyse the indications for CSs in Group 1 separately.

The findings of this study may help to improve the quality of obstetric care, ultimately improving maternal and foetal outcomes in Latvia.

## 2. Materials and Methods

Data were collected from the electronic medical record system and medical charts of the Riga Maternity Hospital (RMH), the largest childbirth institution in Latvia with more than 5000 deliveries annually. RMH is a secondary-level monodisciplinary perinatal care hospital.

The study was approved by the Research Ethics Committee of Rīga Stradiņš University (Document No. 6-1/02/65, 27 February 2020). All births in 2019 were included in the study. Prior to analysis, patient records and information were anonymised and de-identified to protect confidentiality.

All births were classified using the Robson ten-group classification system, including subgroups for Groups 2, 4, and 5 (Table 1) used in the RMH since 2011.

Nulliparous women were defined as women with no previous delivery, and multiparous women were defined as women with at least one previous delivery (from 22 weeks, foetus at least 500 g birth weight, all routes of delivery). The women were assigned to Robson groups as described in the flowchart recommended by the WHO [3].

We analysed indications of the CSs performed in Group 1. All medical charts and foetal monitoring records were collected and reviewed by the group of investigators. The lead obstetrician (D.R.) was engaged as an additional expert to help classify questionable cases. If there were 2 or more diagnosed reasons for a CS, the expert determined which was the leading indication for the CS. The analysis of indications for CS in this group was performed using the methodology proposed by J.K. [6]. All indications were divided into three groups: suspected foetal compromise (SFC), dystocia (D), and other (Figure 1). All the operations performed because of suspected foetal compromise when oxytocin was not administered were classified in the SFC group. If oxytocin was administered, the clinical situation was classified as dystocia (D). Based on two variables (cervical dilation (complete/incomplete) and suspected foetal compromise), all women were classified into different dystocia’s subgroups.

Data were analysed using MS EXCEL and IBM SPSS Statistics 23.0 for Windows.

## 3. Results

In total, 5835 women gave birth at the Riga Maternity Hospital in 2019, which constitutes 31.6% [1] of all births in Latvia in 2019. One patient was excluded from the study because of missing data. The overall CS rate at the Riga Maternity Hospital was lower than that in the country in general, being 21.5% rather than 22.0% [1], respectively.

The general characteristics of all delivery patients in the RMH are presented in Table 2.

The largest Robson groups were multiparous and nulliparous women with single-term cephalic pregnancy and spontaneous onset of labour and without previous CSs (Groups 1 and 3), followed by nulliparous and multiparous women with single-term cephalic pregnancy with induced labour or pre-labour CS without previous CSs (Groups 2 and 4) (Table 3).

A high CS rate is seen in Groups 6 and 7 (foetus in the breech position), and Group 8 (multiple pregnancy). These groups are relatively small, and their high in-group CS rates do not have a huge impact on the overall CS rate.

The CS rate is also high in Group 5 (81.4%) and plays a crucial role in the overall CS rate in the hospital (Table 3). Only 29.3% (150/512) of all women in Group 5 had spontaneous onset of labour; 4.5% (23/512) underwent the induction of labour. The majority of all CSs (81.3%, 339/417) in this group were performed before the onset of labour. Only 33.8% (173/512) of women underwent trial of VBAC, and 54.9% (95/173) delivered vaginally.

As Group 5 has the greatest impact on the overall CS rate, for an in-depth analysis it was further subdivided as follows:second delivery after previous CS;women with two or more previous CSs;women with one previous CS and vaginal delivery were subdivided according to the previous mode of delivery—vaginal delivery before CS, CS, and VBAC (Table 4).

The second largest effect on the overall CS rate is for the nulliparous with a single term pregnancy with the foetus in the cephalic position with induced labour or with pre-labour CS (Group 2). CSs in Group 2 were performed mainly after labour induction (75.9%, 198/261); 24.1% (63/261) were pre-labour CSs.

Group 1 was the third largest contributor to the overall CS rate. Dystocia (D) was the main indication for CSs in this group (Table 5): 36.2% of CSs were performed because of suspected foetal compromise (D: SFC) among patients treated with oxytocin because of dystocia, who comprise 5.7% of all CSs. Apgar score < 7 at 5 min after delivery was found in one patient among women undergoing operations because of D: SFC. The second largest group of indications for CSs was SFC. An Apgar score < 7 at 5 min after delivery was set in 2 patients among women who were operated on because of SFC.

## 4. Discussion

Authors have identified the main contributors to the CS rate in the RMH and determined strategies to reduce the CS rate. The CS reducing strategies involve decreasing the CS rate in Group 1, thus reducing it in Group 5, and encouraging the trial of VBAC, thus lowering the number of women with two or more CS in future.

In our results interpretation we have used the recommendations proposed by Robson et al. [7]. The overall CS rate 22.0% in Latvia is higher than other Baltic states—Lithuania and Estonia, 20.4% [8] and 19.4% respectively [9]. Despite the fact, that RMH is secondary level perinatal care unit, the CS rate is lower than in the entire country. 

The main contributors to the CS rate in RMH are not particularly unique, contributing two-thirds of the total CS rate in the country [7]. The main contributors to the CS rate in RMH are Groups 1, 2 and 5, with Group 5 making the greatest contribution [7]. Profound analysis revealed some specific cultural and organisational differences in labour management within the RMH.

Group 5 (single-term cephalic pregnancy with a previous CS) made the greatest contribution to the overall CS rate, as previously reported by Robson and other authors [10]. The CS rate was higher (Table 3) than recommended by Robson et al. 50–60% [7]. The successful VBAC rate in Group 5 was 18.6% in general. 

In our study, we divided Group 5 according to the onset of labour (spontaneous, induced, or pre-labour CS) and according to the number of previous CSs and type of previous delivery/deliveries. This was performed to find out which subgroup provides the greatest contribution to the overall CS rate in Group 5.

The greatest part—69.1%—of Group 5 were women with the second delivery and previous CS. The VBAC rate in this subgroup was 18.1%, even lower than the overall VBAC rate in Group 5 and much lower than the VBAC rates reported in other studies (64.0% to 74.7%) [11,12,13]. Some studies excluded women with an inter-pregnancy interval shorter than 18 months, a baby large for its gestational age, pregnancy complicated by gestational diabetes, and a previous unclassified uterine scar [11]. Medical factors such as diabetes, hypertensive disorders that complicate pregnancy, Bishop score, labour induction, macrosomia, the indication of previous CS (cephalopelvic disproportion), dystocia, and failed induction should be considered as factors affecting the success of VBAC [14], but were not investigated in our study. This could affect the decision about the trial of vaginal birth after one CS, but not to such an extent as to make more than a 40% difference in VBAC rate. There are other reasons for a low VABC rate in groups with one previous CS. In our consideration, one of the reasons could be the belief that a CS is safer for baby and/or mother. Although there are no published studies on the beliefs of Latvian doctors and patients about the safest mode of delivery after a previous CS, we can deduct from other research data that factors such as the predictability, controllability, and comfortability of CS and the belief that CS is safer for baby and/or mother are found to affect the choice for elective CSs [15,16]. This could probably explain the high CS rate in women with one previous CS, which in most cases was performed as pre-labour CS for women who are unwilling or unable to attempt vaginal birth. Unfortunately, this approach expands the obstetric population with two previous CSs without vaginal delivery and increases the CS rate in future. In our opinion, the 100% CS rate in groups with two or more CS can be explained by the existence of an unwritten belief that elective CS after two and more CSs is the safest mode of delivery. This is based on a fear of uterine rupture or other complications and ensuing patient complaints.

Women with previous vaginal delivery, included VBAC, in Group 5 constituted only 12.3% of all women. The CS rate was lower in groups with VBAC after CS, respectively, at 20.8% and 69.2%. The successful VBAC rate is consistent with other findings about predictive factors for successful VBAC trial—previous vaginal delivery, especially previous VBAC [17].

Failure to provide women with evidence-based information about attempted vaginal birth after CS, along with the short- and long-term complications for mother and/or child after a CS, reinforces the myth that pre-labour CS is the best birth type for women with uterine scarring. On the contrary, a recent study about the quality of maternal and newborn care in Latvia found that women who undergo pre-labour CSs are less satisfied with labour (more often no skin-to-skin contact, no early breastfeeding, no rooming in etc.) than women with the spontaneous onset of labour [18].

It is necessary to continue developing national algorithms in Latvia, as the current ones do not define absolute and relative CS indications and do not define standards in every field of perinatal medicine. Society should be more informed about possible complications after CS. Antenatal care providers in outpatient clinics cannot and should not influence the patient’s decision about the mode of delivery. The final decision should be made in the pre-labour CS consultation or at the onset of labour in the delivery ward. It is crucial to encourage women with one previous CS and without any other pregnancy-related complications to wait for the spontaneous onset of labour and attempt vaginal delivery.

Group 1 (nulliparous, single cephalic, ≥37 weeks, in spontaneous labour) is the third largest contributor to the overall CS rate. However, it makes the biggest contribution to the overall CS rate from all deliveries with spontaneous onset. CSs in this group will mostly add to CSs in future Group 5—women with one previous CS. The desirable CS rate in Group 1 is 10% [7]. It was higher in our study—13.1%—and was similar to the CS rate in Western Europe [10] but almost double than that in Northern European countries such as Sweden [19]. Most CSs among nulliparous women with spontaneous labour were performed because of dystocia (D). This outcome correlates with the results of other studies—59.8–83.4% [6,20,21]. A more detailed analysis of indications for CSs showed that operations in this group were mostly performed in women who were treated with oxytocin because of dystocia and later suspected foetal compromise (D: SFC). This shows that after oxytocin was administered in the case of dystocia, suspected foetal compromise later developed, and CS was performed. The second major indication for CS among women assigned to Group 1 was suspected foetal compromise (SFC). This shows that, during normal delivery without dystocia and the need for oxytocin, an abnormal CTG was detected, and CS was performed. A high incidence of CS for suspected foetal compromise does not coincide with the rate of neonatal morbidity—3.8% had an Apgar score lower than 7 after 5 min in this group. This suggests that several aspects in the labour management could have been improved to treat foetal wellbeing and prevent unnecessary CS. First, according to previous studies, oxytocin should be discontinued during the active phase of labour, thus improving the blood supply to the foetus in order to reduce the CS rate [22,23]. Secondly, there should be continuous and consistent improvement in cardiotocography (CTG) interpretation competency based on foetal physiology. The low sensitivity and poor inter- and intra-observer agreement in the interpretation of intrapartum CTG interpretation was proven [24,25]. The recent study showed that the use of CTG in practice is perceived as a team effort rather than an individual task [26]. The abilities of individuals combine with external influences and teamwork within multi-professional teams, resulting in CTG interpretation and subsequent decision making [26]. As CTG is a widely available and minimally invasive method, it will continue to be used to monitor foetal wellbeing. It is likely that smart intrapartum surveillance systems will become universally used in CTG interpretation, as studies have shown that this approach reduces the CS rate in nulliparous women with term cephalic pregnancy [27].

The second largest contributor to the overall rate of CS was Group 2 (nulliparous, single cephalic, ≥37 weeks induced labour or pre-labour CS), accounting for 20.8% of all CSs in the RMH. Within Group 2, 75.8% (198/261) were women with induced labour. The CS rate for nulliparous women with induced labour varied in previous studies from 10.2% to 38.7% [28,29,30] depending on gestational age at the time of induction, labour management, national guidelines and traditions, and other factors. The CS rate in the RMH for this subgroup was not outstanding (24.9%), while nulliparous women with term singleton pregnancy and spontaneous labour onset or induced labour made the same absolute contribution to the overall CS rate - 3.4% each. As the number of CSs in the induced labour group has a significant impact on the total number of CSs, and as the size of this group is currently increasing [31,32], the choice of the best strategy for successful induction of labour is an important field of research. According to the current evidence and clinical recommendations of the Latvian Association of Gynaecologists and Obstetricians, women with risk factors such as hypertension, diabetes, and foetal growth retardation, etc., are recommended to deliver before 40 weeks of gestation to reduce the risk of foetal demise [33].

### Strengths and Limitations

There are many strengths to our study. First, it should be emphasised that the study was conducted in the biggest childbirth institution in Latvia, accounting for more than 30% of all deliveries in the country. Therefore, the sample size of our study is considerably representative of the Latvian population. Second, data were collected, and deliveries classified by a group of experts using Robson classification. For the assessment of data quality, the WHO recommends several criteria [3] that are fulfilled in our study: only one woman was not included in study because of missing data about previous delivery type, and Group 9 accounts for less than 1% of the whole population in our study - 0.1% with a 100% CS rate.

A possible limitation of our study was that we did not analyse other data (inter-pregnancy interval shorter than 18 months, pregnancy complicated by gestational diabetes or hypertensive disorders, etc.) related to pregnancy and previous deliveries, which may affect the decision about the mode of delivery. In future, studies about patients’ and doctors’ beliefs and worries about mode of delivery after CS in Latvia should be carried out to understand the leading factors (medical, psychoemotional etc.) for high pre-labour rate in Group 5.

## 5. Conclusions

The main contributors to the CS rate in the RMH are Groups 1, 2 and 5, with Group 5 being the most regular contributor. Based on our analysis of Groups 1 and 5, the RMH CS rate reduction strategy first, should focus on decreasing the CS in nulliparous women. It is important to reduce the number of CSs in Group 1 by improving the protocol of labour dystocia management, especially when signs of foetal compromise develop after prescribing oxytocin. Second, it is important to encourage women with one previous CS to attempt VBAC. Both sets of goals would help to reduce the number of CSs in the obstetric population with one or more previous CSs, and this should be emphasised in educating professionals and society. To paraphrase Edwin Cragin’s famous century-old maxim, ‘once cesarean section, always cesarean section’ [34], we advocate a philosophy of ‘no first cesarean section, no next cesarean section’.

## Figures and Tables

**Figure 1 jcm-12-06426-f001:**
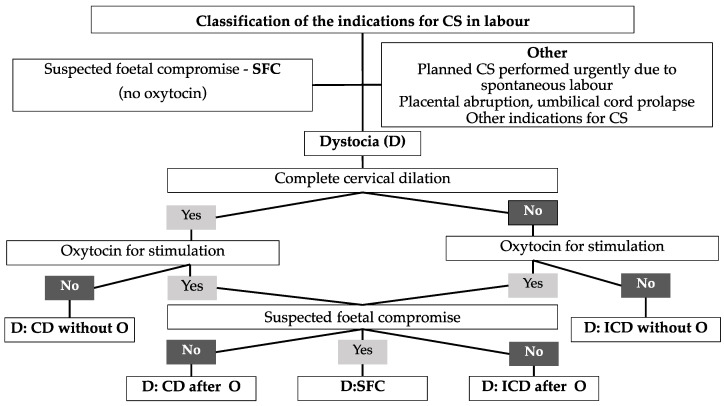
Classification of the indications for CS. D: CD without O—dystocia, complete cervical dilation without oxytocin; D: CF after O—dystocia, complete cervical dilation after oxytocin; D: SFC—dystocia with oxytocin for stimulation and later suspected foetal compromise; D: ICD after O—dystocia, incomplete cervical dilation after oxytocin; D: ICD without O—dystocia, incomplete cervical dilation without oxytocin.

**Table 1 jcm-12-06426-t001:** Robson Ten-Group Classification System with subgroups used in RMH.

Group	Description
Group 1	Nulliparous, single cephalic, ≥37 weeks, in spontaneous labour
Group 2	Nulliparous, single cephalic, ≥37 weeks
Subgroup 2a	with induced labour
Subgroup 2b	with pre-labour CS
Group 3	Multiparous (without previous CS), single cephalic, ≥37 weeks, in spontaneous labour
Group 4	Multiparous (without previous CS), single cephalic, ≥37 weeks
Subgroup 4a	with induced labour
Subgroup 4b	with pre-labour CS
Group 5	Previous CS, single cephalic, ≥37 weeks
Subgroup 5a	spontaneous onset of labour
Subgroup 5b	with induced labour
Subgroup 5c	with pre-labour CS
Group 6	All nulliparous breeches
Group 7	All multiparous breeches (including previous CS)
Group 8	All multiple pregnancies (including previous CS)
Group 9	All transverse/oblique lies (including previous CS)
Group 10	All preterm single cephalic, including previous CS

**Table 2 jcm-12-06426-t002:** General characteristics of all delivery patients in RMH.

Characteristics			Frequency (*n*)	Percentage, %
Maternal age (years)	<20		112	1.9%
	20–29		2225	38.1%
	30–34		2100	36.0%
	35–39		1105	18.9%
	≥40		292	5.0%
Parity	1		2613	44.8%
	>1		3221	55.2%
Previous CS	No		5248	90.0%
	Yes		586	10.0%
Number of foetuses	Single		5741	98.4%
	Multiple		93	1.6%
Mode of birth	Vaginal		4322	74.1%
	Operative vaginal		259	4.4%
	CS		1253	21.5%
Gestational age	< 32^+0^		55	0.9%
	32^+0^ – 36^+6^		262	4.5%
	37^+0^ – 38^+6^		897	15.4%
	39^+0^ – 40^+6^		3291	56.4%
	≥41^+0^ + 0		1329	22.8%
Birth weight	<2500 g		262	4.4%
	2500–2990 g		624	10.5%
	3000–3990 g		3984	67.2%
	≥4000 g		1052	17.7%
	Missing data		5	0.1%
Apgar score <7 at 5′			45	0.8%

**Table 3 jcm-12-06426-t003:** The proportion of each Robson group, size of the group (%), CS (%), and their relative and absolute contribution to the overall CS rate.

Group	Number of CSs in Group	Number of Women in the Group	Group Size, %	Group CS Rate, %	Absolute Group Contribution to the Overall CS Rate, % *	Relative Contribution of the Group to the Overall Rate to CS, % **
1	196	1499	25.7%	13.1%	3.4%	15.6%
2	261	858	14.7%	30.4%	4.5%	20.8%
2a	198	795	13.6%	24.9%	3.4%	15.8%
2b	63	63	1.1%	100.0%	1.1%	5.0%
3	48	1880	32.2%	2.6%	0.8%	3.8%
4	44	564	9.7%	7.8%	0.8%	3.5%
4a	30	550	9.4%	5.5%	0.5%	2.4%
4b	14	14	0.2%	100.0%	0.2%	1.1%
5	417	512	8.8%	81.4%	7.1%	33.3%
5a	67	150	2.6%	44.7%	1.1%	5.3%
5b	11	23	0.4%	47.8%	0.2%	0.9%
5c	339	339	5.8%	100.0%	5.8%	27.1%
6	94	107	1.8%	87.9%	1.6%	7.5%
7	39	63	1.1%	61.9%	0.7%	3.1%
8	60	93	1.6%	64.5%	1.0%	4.8%
9	6	6	0.1%	100.0%	0.1%	0.5%
10	88	252	4.3%	34.9%	1.5%	7.0%
Total	1253	5834	100.0%	NA	21.5%	100.0%

* Absolute contribution (%) = number of CSs in the group/total number of women delivered in the hospital × 100. ** Relative contribution (%) = number of CSs in the group/total number of CSs in the hospital × 100.

**Table 4 jcm-12-06426-t004:** Group 5 according to previous type of delivery.

Subgroup According to Previous Type of the Delivery	Total (*n*)	Vaginal Delivery (*n*, (%))	CS before the Onset of Labour (% from CSs in Subgroup)	CS after the Onset of Labour (% from CS in Subgroup)	CSs (*n*, (%))	Relative Contribution of the Group to the Overall Rate of CS, %
One previous CS—2nd delivery	354/512	64	229	61	290	23.1%
	(69.1%)	(18.1%)	(79.0%)	(21.0%)	(81.9%)	
Vaginal delivery before CS	39/512	12	21	6	27	2.2%
	(7.6%)	(30.8%)	(77.8%)	(22.2%)	(69.2%)	
CS and VBAC *	24/512	19	3	2	5	0.4%
	(4.7%)	(79.2%)	(60.0%)	(40.0%)	(20.8%)	
Two or more previous CSs	95/512	-	89	6	95	7.6%
	(18.6%)		(93.7%)	(6.3%)	(100.0%)	
Total:	512	95	342	75	417	33.3%
		(18.6%)	(82.0%)	(18.0%)	(81.4%)	

* Vaginal birth after CS.

**Table 5 jcm-12-06426-t005:** Indications for CSs in Group 1.

No	Subgroup According to Previous Type of the Delivery	Absolute Contribution to the CS Rate in Group 1, % (*n*/all CS in Group 1)	Relative Contribution to the Overall CS rate, %	Apgar Score less than 7 at 5 min (*n*, % from all SUBGROUP)
**1**	**Suspected foetal compromise** (**SFC**)	**26.5%** (**52/196**)	**4.2%**	**2** (**3.8%**)
**2**	**Dystocia** (**D**)	**69.4%** (**136/196**)	**10.9%**	**2** (**1.5%**)
2.1.	D: CD without O	2.6% (5/196)	0.4%	0
2.2.	D: CD after O	11.2% (22/196)	1.8%	0
2.3.	D: ICD without O	6.1% (12/196)	1.0%	0
2.4.	D: ICD after O	13.3% (26/196)	2.1%	1 (3.8%)
2.5.	D: SFC	36.2% (71/196)	5.7%	1 (1.4%)
**3**	**Other**	**4.1%** (**8/196**)	**0.6%**	**0**

## Data Availability

The datasets generated are available from the corresponding author upon reasonable request.

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
