# Peer review of "In-Depth Analysis of Caesarean Section Rate in the Largest Secondary Care-Level Maternity Hospital in Latvia"

_jcm, 2023, doi:10.3390/jcm12196426_

Round 1

Reviewer 1 Report

This is a very interesting quality improvement study. In modern obstetrics, the frequency of caesarean sections appears to be the most important indicator of the quality of health care. Your study shows, that the different groups of the Robson classification can be analysed in even more detail. The study thus offers new findings for clinical practice. Similar quality-improvement study related to the reduction of caesarean sections was published in IJGO in 2020, it would be interesting to confront your results with this study. 

Reviewer 2 Report

The present study is a Robson classification of the deliveries in a single center during 2019. 

Comments:

1. Robson (TGC) classification was intended to measure trends; i. why did the authors choose 2019 ?; ii. a trend presentation of a few years could improve the report and substantiate the conclusions of only 5000 deliveries.

2. How was the diagnosis for the CS retrieved from the records in case of multiple diagnoses; e.g. dystocia and fetal distress. Did the authors used both diagnoses ? first recorded? the data base had " MAIN DX" section ?

3. An additional analysis is suggested to mitigate the influence of other causes for CD (  maternal background disease; pregnancy complications) at least for a subgroup that this information is available. Since it is a single center it should clarify the specific influence of the RG. 

4. The aim " identify specific goals ..." is not related to the conclusion. The authors should mention that this is a descriptive local study. 

Reviewer 3 Report

The authors present a manuscript which aims to identify the factors related with increased cesarean delivery rates in Latvia. Although the topic does not appear to be so original and authentic, this research contributes to literature by clarifying the obstetric goals for reducing cesarean delivery rates in a developed European country where primary health services can be used for achieving these goals. Moreover, the topic of this research addresses a very basic yet usually neglected concept of "coping with increased cesarean delivery rates" which has significance for all developing countries located worldwide. The similar research in literature mainly focuses on the preventive measures for increased cesarean delivery rates while this manuscript aims to specify the factors related with these increased rates based on Robson classification. The methodology of the study has been set up as efficiently as possible and the conclusions comply with the evidence presented by the statistical findings. Additionally, both the conclusions and evidence are all related with the main question and hypothesis of the manuscript and all the references are relevant and up-to-date. The tables are also well drawn and demonstrate the required data. 

Round 2

Reviewer 2 Report

The authors still did not reply why 2019  alone was reported and no trends shown.

The abstract and the title were not modified according to the descriptive study design ( rather than interventional)
